# Direct observation of phase transitions in truncated tetrahedral microparticles under quasi-2D confinement

David Doan [1,2], John Kulikowski[1,2] & X. Wendy Gu [1] ✉

Colloidal crystals are used to understand fundamentals of atomic rearrangements in condensed matter and build complex metamaterials with unique functionalities. Simulations predict a multitude of self-assembled crystal structures from anisotropic colloids, but these shapes have been challenging to fabricate. Here, we use two-photon lithography to fabricate Archimedean truncated tetrahedrons and self-assemble them under quasi-2D confinement. These particles self-assemble into a hexagonal phase under an in-plane gravitational potential. Under additional gravitational potential, the hexagonal phase transitions into a quasi-diamond two-unit basis. In-situ imaging reveal this phase transition is initiated by an out-of-plane rotation of a particle at a crystalline defect and causes a chain reaction of neighboring particle rotations. Our results provide a framework of studying different structures from hard-particle self-assembly and demonstrates the ability to use confinement to induce unusual phases.

Colloidal particles can self-assemble into ordered crystals with extraordinary nano and mesoscale complexity[1] and unique optical, electronic, and magnetic properties[2,3]. These emergent properties depend on the properties of the constituent particle and the crystal phase of the final ordered structures. The phase behavior of self-assembled colloidal structures depend on a variety of factors, such as shape, surface interactions, and external fields[1]. Two-dimensional (2D), hard-particle colloidal systems are of interest because they are entropically driven, and their final assembly state solely depends on the shape and packing fraction of the particles. Previous computational[4] and experimental studies have shown interesting crystallization behavior (from liquid to solid) and crystal structures in 2D systems consisting of spherical colloids[5,6], ellipses[7,8], rods and rectangles[9–11], squares[12,13], triangles[14,15], and hexagons[16]. Hard or nearly hard spheres are commonly observed to form face-centered cubic structures. Complex three-dimensional (3D) structures such as diamond, space-filling polyhedral packing, and porous lattices have been formed by using patchy DNA interactions, shape-dependent entropic forces, or magnetic, gravitational, and capillary forces[17–22]. A wide range of hard-particle 3D assemblies have been extensively predicted in simulation[23], but are challenging to experimentally achieve and image.

Colloidal crystals are often described as programmable materials[24] but typically form static structures that cannot be reconfigured into different crystals once assembled, or can only be dissembled and re-assembled into the same structure[25]. The ability to directly switch between distinct crystal structures is analogous to solid-solid phase transitions in atomic matter and has previously been studied in different colloidal self-assembled systems[26]. Phase transition kinetics in soft spherical colloids have been previous studied under electric fields[27]. Phase transitions that maintain crystal symmetry can be induced in DNA-functionalized nanoparticle superlattices by inserting additional nanoparticles or DNA linkers[28,29]. Phase transitions have also been investigated with hard-particle spherical colloids[30,31]. The majority of the studies have been in 2D systems that require complicated external fields. 3D phase transitions (e.g. from FCC phase to AuCu phase) have been observed using X-ray scattering[32–34] and confocal techniques[35–37]. In hard-particle systems, one strategy is to change the colloidal shape to an anisotropic, or

[1]Department of Mechanical Engineering, Stanford University, Stanford, CA 94305, USA. [2]These authors contributed equally: David Doan, John Kulikowski. ✉e-mail: xwgu@stanford.edu

higher order polygon, which can result in complex phase behavior such as crystal-crystal or solid-solid phase transitions. Colloidal squares[13] that were assembled in 2D have shown complex phase behavior as a function of packing fraction. Superballs[38] that have been assembled in 3D have also shown solid-solid phase transition under different osmotic pressures. However, these superball assemblies show similar phase behavior as those found in 2D systems[12,13]. In addition to shape change, an external potential, such as confinement or boundary conditions, can also play a large role in the possible accessible crystal phases[39–42].

In general, these previous hard-particle phase transitions lack complex phase transformations, such as those in which the "atoms" change coordination number, or transform between crystal lattice systems. Further progress in this field of work could lead to metamaterials with rapidly switchable properties and functional structures. Elucidating the kinetics of colloidal phase transitions could also provide understanding of solid-solid phase transitions in atomic solids, which remain controversial even for elemental materials due to the challenges of observing dynamic behavior at the atomic scale[43,44]. The advantage of these colloidal systems is that they can be imaged at a spatial and temporal resolution that cannot be achieved in real atomic systems, even with state-of-the-art experimental tools such as transmission electron microscopy[45] or ultrafast X-ray diffraction[46–48].

In this work, we assemble lithographed Archimedean truncated tetrahedrons (ATT) at an interface to achieve quasi-2D confinement. This strategy takes advantage of the high dependence of shape on the phase behavior of the final assembled state, in addition to subjecting the system to a boundary condition that has been previously shown to induce rich phase behavior in polygons. The Archimedean truncated tetrahedron was chosen because simulations of truncated tetrahedrons in 3D show phase behavior that is highly dependent on the truncation parameter, $t$, (see Methods) with crystalline structures that are analogous to important atomic crystals. For example, ATTs ($t = 2/3$), which have four regular hexagonal faces and four regular triangular faces with all the same edge lengths, are predicted to form diamond structure at lower packing densities ($\approx 0.6$), and α-arsenic at the highest packing density ($\approx 1$)[49]. Simulations of truncated polyhedrons (i.e. cubes, octahedrons) constrained to a 2D plane have been previously explored[41], but truncated tetrahedrons have yet to be studied. ATTs have also been studied in simulation under spherical and wall confinements[40], but their 2D behavior on a surface was not further explored. Although the behavior of these types of polyhedrons are of interest, the main experimental limitation is the ability to fabricate such geometries with low dispersity in size and shape.

## Results

### Hexagonal phase induced at low tilt angles
To overcome the synthetic challenges of forming polyhedral particles with low dispersity, two-photon lithography is used to fabricate ATT microparticles with a side length of 3.5 μm (Fig. 1a). Approximately 50,000 particles are fabricated with ≤ 5% variation in particle size[50]. Other tetrahedral particles, such as regular tetrahedrons ($t = 0$) and truncated tetrahedrons ($t = 7/10$), are also easily fabricated using this method (see Suppl. Fig. 1). After fabrication, the particles are dispersed in water and deposited in a well plate for assembly. Initially, the particles randomly sediment on the substrate and are dispersed across the substrate with low packing density. We observe that the particles are generally oriented with a hexagonal side facing the substrate, with a triangular face pointing upward, referred to as the 'upright' position. This is due to the center of gravity of the particle being weighed towards the hexagonal face.

The substrate is then tilted to apply an in-plane ($x/y$ direction) gravitational potential field. This gravitational field leads to an induced osmotic pressure and density gradient along the direction of tilt. After several days ($\approx 144$ h), the particles aggregate to one side of the well

plate which increases the local packing density and causes ordered regions to form.

At a $\approx 5°$ tilt angle, the ATTs form a hexagonal phase (Fig. 1b). In this phase, the particles, which are oriented with their hexagonal face in contact with the substrate, have 6 nearest neighbors. For this geometry to form, three of the triangular faces of each particle are in face-to-face contact with the hexagonal faces of its neighboring particles as shown in Fig. 1c. This effectively "locks" the particle into place by preventing the neighboring particle from moving in the $z$-direction. The grain size and rotational order is analyzed using a bond orientational order parameter that accounts for the 6-fold symmetry of the assembled structures (see Methods)[49]. This order parameter is represented as colors in Fig. 1d. Using this analysis, grains are identified as particles with the same color and found to be ≈30 μm or 30–40 particles in size. Grains are separated by vacancies (missing particles) and point defects (disordered particles). The spatial pair distribution function, $g(r)$, is used to quantify the translational packing order (Fig. 1e). The $g(r)$ plot shows a first peak at ≈6.6 μm followed by a double peak. This is indicative of a hexagonal phase, which has been observed in 2D assemblies of spherical colloids on a surface[51]. The corresponding Fourier transform shows bright spots in a hexagonal geometry, which is indicative of a hexagonal phase.

### Quasi-diamond phase induced at higher tilt angles
The ATTs are then tilted by an additional ≈5° to an angle of ≈10° and allowed to assemble over 48 h. This results in a phase that is drastically different than the previous hexagonal phase. This is reflected in the optical images as triangular shapes that are arranged with 3 nearest neighbors. Because of the drastically different particle shape in the optical microscope, the particle orientations are elucidated by using confocal imaging at different $z$-planes (Fig. 2a–c). These images show alternating triangular and hexagonal faces near the substrate. As we focus away from the substrate, the triangular faces become larger, and the hexagonal faces become more triangular (Fig. 2b, c). This demonstrates that the ATTs form a nearly space-filling structure made up of a two-particle unit cell that consists of one 'upright' facing ATT and one 'upside-down' facing ATT. This structure, which we refer to as quasi-diamond, is equivalent to a two-atom basis in diamond cubic structure. This has been predicted to form from ATTs that self-assemble under an entropic driving force at packing fractions above 0.50[49].

The bond orientational order parameter is calculated for the quasi-diamond structure (Fig. 2e). These images are obtained at a focal plane near the center of the particle which cause the ATTs to appear as triangular shapes under these imaging conditions. Grains are shown as regions of alternating colors and are approximately half the size ( ≈ 20 um) of the hexagonal grains. The spatial pair distribution function, $g(r)$, and corresponding Fourier transform show a first peak at ≈4.4 um, a weaker second peak, and no additional peaks (Fig. 2f). This indicates the formation of a 3-fold symmetric phase with short range order, small grains, and a higher density of defects as compared to the hexagonal phase. This $g(r)$ and bond orientational order is similar to that of self-assembled triangular plates that form 3-fold, 2D structures[14], as well as self-assembled, two-photon lithographed regular tetrahedrons (see Supplementary Fig. 2).

Here, we consider the thermodynamics of self-assembly. For hard particle systems, this behavior can be examined through the lens of entropy maximization[52]. In these systems, self-assembly is dominated by an entropic driving force due to the gain in free volume when the particles form an ordered arrangement. Generally, the free volume is maximized when particles are in face-to-face arrangements[53]. For the disordered (right after deposition) to hexagonal phase transition, the increase in face-to-face area is due to the contact of the three of the triangular faces of the ATT with the hexagonal faces of its neighboring particles. The hexagonal to quasi-diamond phase transition results in

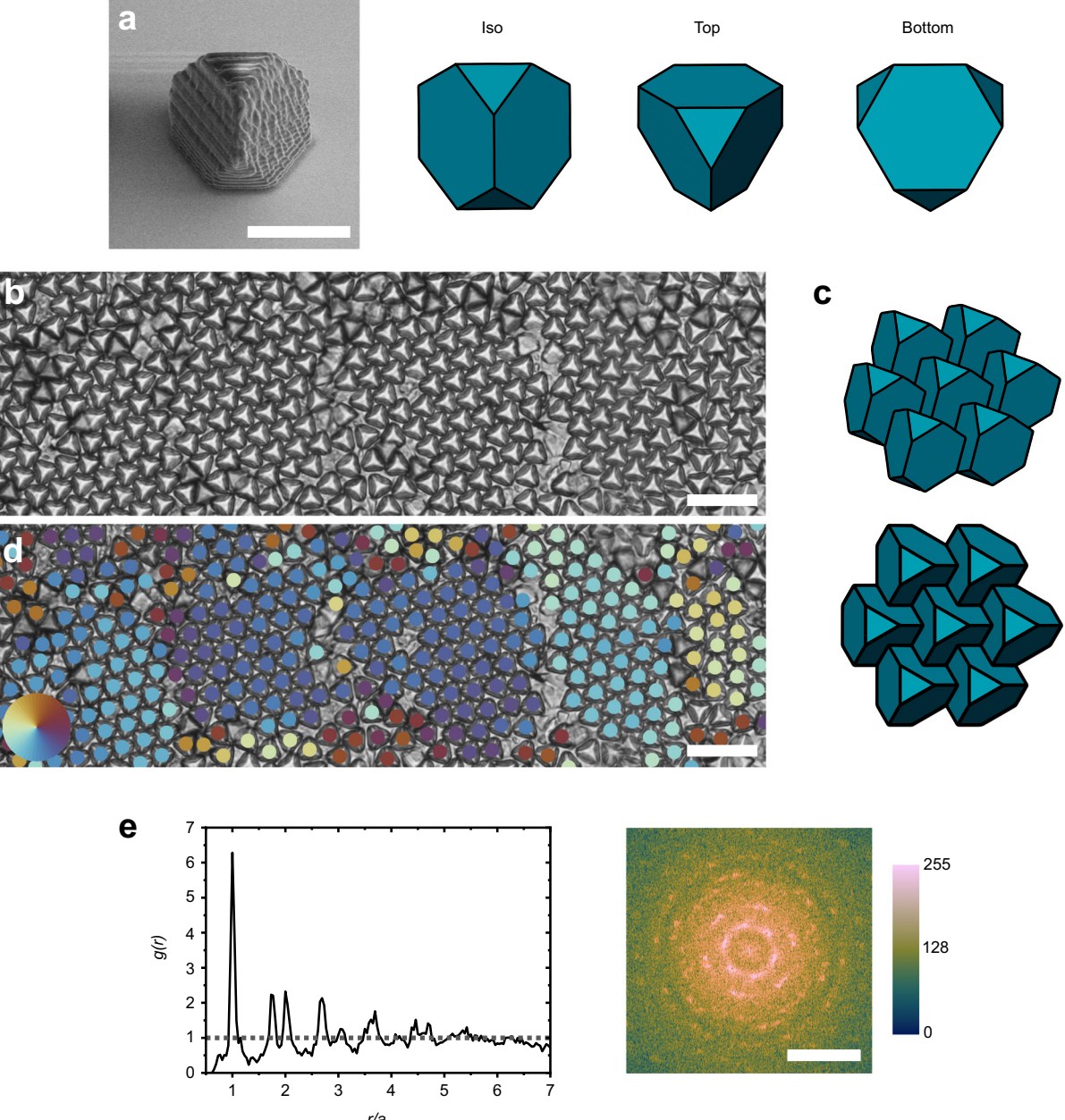

**Fig. 1 | Hexagonal phase. a** SEM image and 3D model of ATT (isometric, top, and bottom view). Scale bar is 5 μm. **b** Optical image of self-assembled hexagonal structure. Scale bar is 20 μm. **c** 3D model of self-assembled structure (isometric and top view). **d** Bond orientational order parameter of the particles represented as different colors. Particles with similar colors have similar rotational orientation. Particles with opposite colors on the color wheel are rotated by 30°. Scale bar is 20 μm. **e** Pair distribution function, $g(r)$ and Fourier transform of image **b**. Scale bar is 0.5 μm⁻¹. Color bar corresponds to 8-bit grayscale.

further gains in entropy because the face-to-face contact increases by >100%. The free volume change between hexagonal and quasi-diamond phase can also be computed directly. The total volume of the system is considered as an $x$-$y$ box that fits $N$ particles, with a $z$-height of one particle unit. Using this as the total volume accessible to the particles, the hexagonal phase has a maximum packing fraction of ≈64% while the quasi-diamond structure is a nearly space-filling structure at ≈99%. Therefore, the relative change in volume density between hexagonal phase and the quasi-diamond phase is ≈50%. This indicates that there is a large driving force towards the quasi-diamond phase from the hexagonal phase. Other truncated tetrahedron ($t = 7/10$) form even smaller quasi-diamond grains because of a lower change in free volume and lower driving force for self-assembly (see Supplementary Fig. 3).

## Free energy single-cell occupancy model

An approximate single-cell occupancy model is used to estimate free energy ($F$) as a function of packing fraction ($\phi$) (Fig. 3a)[54,55]. This model is one of the several methods to analytically calculate the free energy of a hard-particle system and has only been used to model hard spheres in different phases[56–60]. Other similar cell methods have also been developed to calculate the densest packing states of polygons, and subsequently, the free volume of certain packing structures[61,62]. Cell models use the phase of interest (for spheres, face-centered cubic or hexagonal closed-packed) at the highest packing fraction and partitions each particle center inside Voronoi polyhedrons, known as cells. For spheres packed in a face-centered cubic or hexagonal closed-packed phase, the corresponding Voronoi polyhedron would be a dodecahedron. This model assumes that each particle is constrained within its own cell and

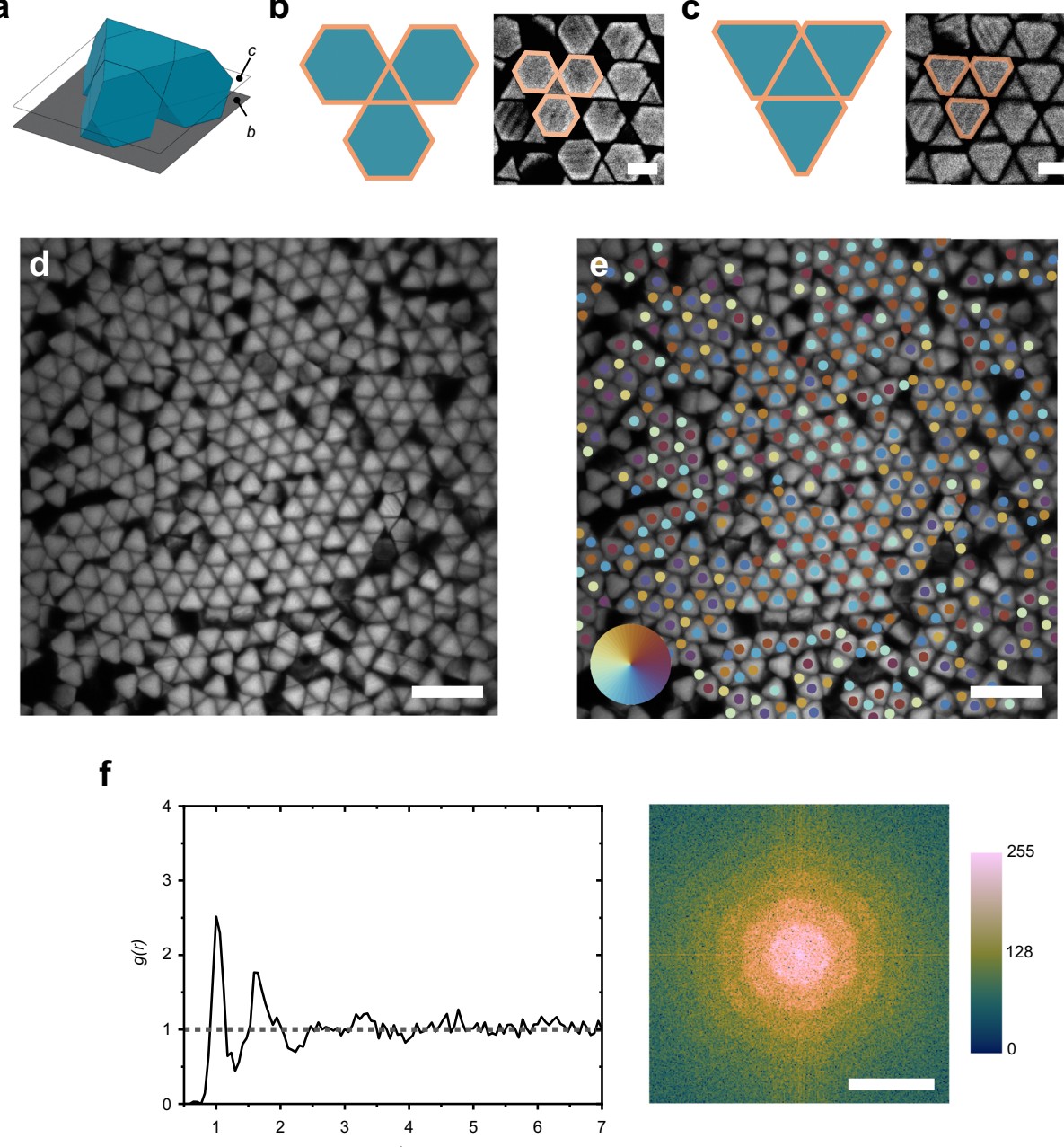

**Fig. 2 | Quasi-diamond phase. a** 3D model of self-assembled structure. The planes that correspond to images b and c are marked. **b, c** Confocal images and 2D models of the same quasi-diamond structure at different focal planes. **b** is focused at the substrate (gray) and **c** is focused at the middle of the particle. The peach outline shows the analogous geometry between the model and the confocal images. Scale bars are 5 μm. **d** Confocal image of a large region of the sample. Scale bar is 20 μm.

**e** The bond orientational order parameter of the particles is represented as different colors. Adjacent particles with opposite colors on the color wheel indicate the quasi-diamond structure (e.g., blue and brown). Scale bar is 20 μm. **f** Pair distribution function, $g(r)$, and Fourier transform of image **d**. Scale bar is 0.5 μm⁻¹. Color bar corresponds to 8-bit grayscale.

can only access the volume associated with its own cell. The Voronoi cell can then be scaled equally in 3D to decrease the packing fraction, and effectively increase the volume accessible to each particle. The total accessible volume of each cell can then be used to estimate the free energy of the system as a function of packing fraction, $\phi$:

$$F(\phi)/(Nk_BT) \approx \ln\left(V_{\text{free}}(\phi)\right) \qquad (1)$$

The single-cell occupancy model is most accurate at higher packing fractions when the assumptions are more likely to be satisfied. For lower packing fractions, this error is associated with a "communal

entropy"[59]. For our system, which consists of polyhedron shapes and is quasi-2D, a slightly different procedure is taken. Instead of constructing Voronoi cells and dilating in 3D, self-similar cells are constructed around each particle from their phase (either hexagonal or quasi-diamond) and then dilated in two dimensions (Fig. 3a and Supplementary Fig. 4). That is, the single-cell is constructed to be in a shape of an ATT, and then stretched only in the x-y dimension. Because of the shape of the cell, the volume of the single cell, $V_{\text{cell}}$, can be calculated analytically for different dilated states. The accessible free volume, $V_{\text{free}}$, is then taken as $V_{\text{cell}} - V_{\text{ATT}}$, where $V_{\text{cell}}$ varies as a function of the dilation and $V_{\text{ATT}}$ is constant.

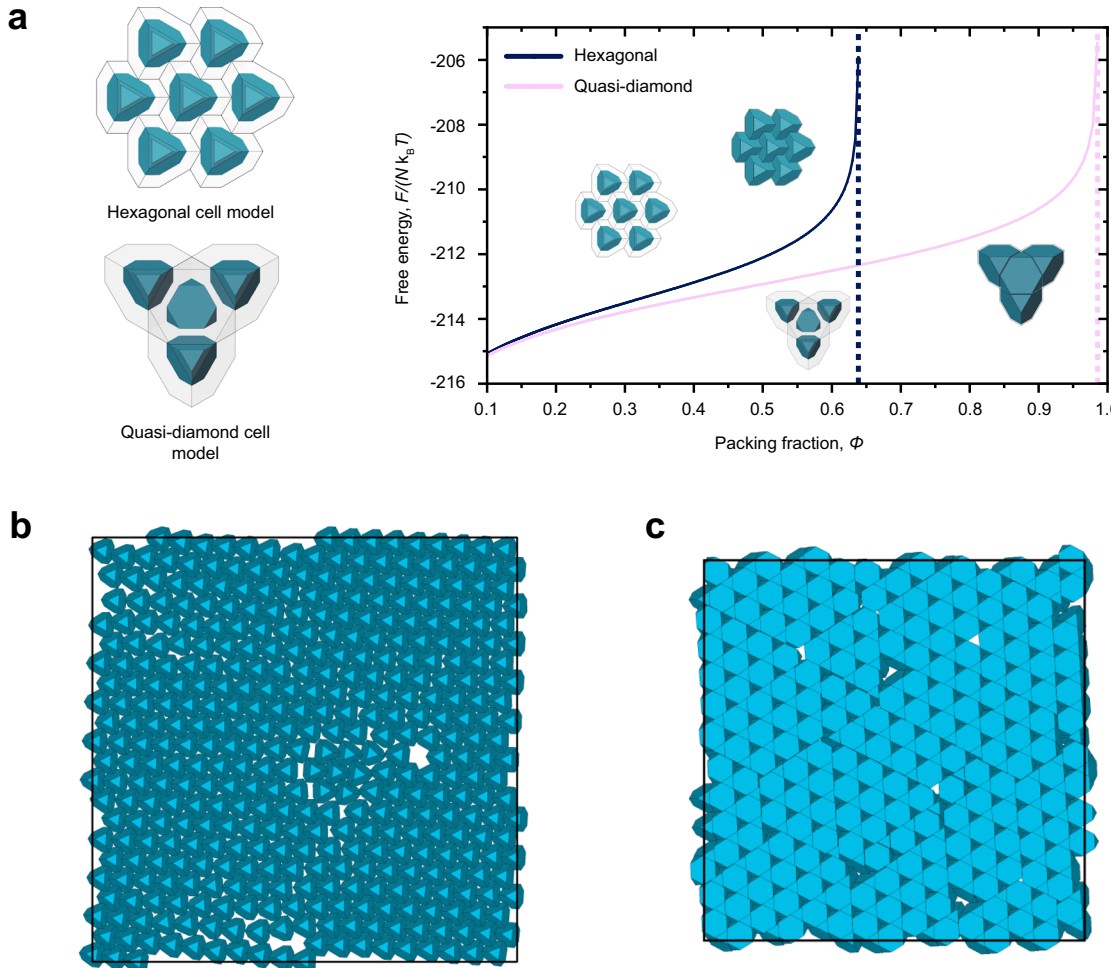

**Fig. 3 | Analytical model and hard particle Monte Carlo simulation. a** Geometric models of self-assembled particles within volumetric cells and the resulting single-cell occupancy free energy calculations as a function of packing fraction of the hexagonal state (navy blue) and the quasi-diamond state (pink). **b** Monte Carlo simulation of ATTs constrained to a 2D plane and laterally compressed results in the formation of a hexagonal phase. **c** Monte Carlo simulation of ATTs after removal of the 2D constraint. Continued lateral compression leads to the formation of the quasi-diamond phase.

We find that the quasi-diamond phase has a slightly lower free energy than the hexagonal phase at all packing fractions. However, this difference becomes larger when packing fraction increases, especially when it approaches the maximum hexagonal packing density. As the packing fraction reaches the theoretical maximum packing fraction of its phase, the free energy approaches infinity. The quasi-diamond phase is not accessible until there is sufficient energy to overcome the thermodynamic or kinetic barrier of the transition from the hexagonal to the quasi-diamond phase. We attribute this barrier to the effects of the quasi-2D confinement, which prevents particle out-of-plane rotation. A thermodynamic barrier exists due to the gravitational potential at low tilt angles. A hexagonal to quasi-diamond phase transition would require 50% of the particles to flip from an 'upright' orientation to an 'upside-down' orientation. The flipping of a particle corresponds to an increase in gravitational potential energy. The gravitational energy required to flip a particle upside down is calculated as $\Delta E = mg\Delta h$, where $\Delta h$ is the change in height of the center of gravity of the particle in its "upright" (hexagonal face is adjacent to the substrate) vs "upside down" (triangular face is adjacent to the substrate) position. The energy needed to flip the particles is $\approx 10\ k_B T$, which suggests that it is very unlikely that out of plane rotation can occur spontaneously at room temperature without external energy input (see Suppl. Movie 1). In addition, a kinetic barrier also exists due to the free volume required to mediate

the rotation of a particle out-of-plane from its locked hexagonal phase to the quasi-diamond phase.

We can test the hypothesis that the phase transition energy barrier is related to an out-of-plane particle rotation by using hard particle Monte Carlo simulations (Fig. 3b, c and Suppl. Movie 2). ATTs are first confined to a 2D plane such that the particles cannot rotate out-of-plane and can only move in the *x-y* directions. The particles are laterally compressed until they approach the maximum theoretical packing fraction. This leads to the formation of a hexagonal structure, with some defects (Fig. 3b) as seen in experiments. This 2D constraint is then removed, which allows out-of-plane rotation. Once this constraint is removed, the particles almost immediately form the quasi-diamond phase under continued lateral compression (Fig. 3c).

In-situ optical microscopy reveals the kinetics of phase transition. First, a hexagonal sample is assembled through a small tilt angle ($\approx 5°$) as previously described. This sample is then tilted by an additional $\approx 5-10°$ and moved to the microscope stage. By the time that imaging begins ($\approx 10$ min after tilting to $\approx 5-10°$), many hexagonal regions have already transformed to quasi-diamond. However, the transition of the remaining hexagonal phase can be observed.

**Phase transitions are initiated by defects**
These in-situ experiments show that the phase transition is mediated by defects and that these defects allow for out-of-plane particle

rotation. Figure 4 shows specific instances of these defect induced phase transitions (see Supplementary Movies 3 and 4). Figure 4a–h show a vacancy mediated phase transition. Initially, a hexagonal grain is surrounded by the quasi-diamond phase with a vacancy present near the phase boundary. The hexagonal particle adjacent to the vacancy rotates out-of-plane and is 'upside-down', in which a triangular face is adjacent to the substrate (Fig. 4i). This leads to a chain reaction in which the next particle rotates and transforms, and then the next particle, until the hexagonal phase has fully transformed into a quasi-diamond phase. The presence of the vacancy seems to facilitate an out-of-plane rotation of the ATT particle by providing the free volume to accommodate an out-of-plane rotation.

Direct observation of a phase transition is also observed at an anti-phase boundary between two hexagonal grains which have particles oriented in different directions (Fig. 4j–m). The two rows above (green hexagonal grain) and the row below (peach hexagonal grain) the anti-grain boundary (army-green dashed line) undergo a phase transition to the quasi-diamond phase. The transition occurs rapidly for half the particles, while the remaining particles in these rows begin to rotate into a transition state (begin flipping out-of-plane, Fig. 4k). This is followed by the transition of the remaining particles at the anti-phase boundary into the quasi-diamond phase (Fig. 4l) and further growth of the quasi-diamond phase until two smaller, isolated hexagonal grains remain (Fig. 4m).

In Fig. 4, the propagation of the phase transformations is perpendicular to the tilt direction with the transition proceeding in a linear direction. However, this is not always the case. Supplementary Movie 5 shows a randomly proceeding phase transition for ATTs. Multiple particle flipping events occur and propagate inward, transforming the structure from a hexagonal to quasi-diamond phase. This suggests that while a mechanical driving force is necessary to induce the phase transition, entropy does in fact play a role. Likely, the phase transition is driven by a combination of mechanical and thermodynamic driving forces.

Without these defects, the hexagonal to quasi-diamond phase transitions is kinetically improbable: an ATT particle would need to escape from its "locked" hexagonal configuration, and then rotate out-of-plane. This kinetic pathway is unlikely, given that the "locked" hexagonal configuration geometrically prevents out-of-plane motion. However, once a particle successfully rotates into a quasi-diamond phase, the local packing density of the particles around it is lowered because the quasi-diamond phase is ≈50% denser than the hexagonal phase. This allows neighboring particles to also have more free space to rotate out-of-plane and continue propagating the phase transition. This type of defect mediated transition is also seen in the simulations, right after the removal of the 2D constraint (see Supplementary Movie 2). By analyzing a hexagonal to quasi-diamond phase transition, the phase transition rate was found to follow Avrami's solid-solid phase kinetic theory in 2D[63,64] (see Supplementary Fig. 5).

In summary, we have assembled Archimedean truncated tetrahedrons under quasi-2D confinement and shown a hexagonal phase that has not been previously reported in literature for this shape. We directly imaged a phase transition from a hexagonal phase, which has 6 nearest neighbors, into a quasi-diamond phase, which has 3 nearest neighbors. We determined the thermodynamics and kinetic mechanism of this phase transition using analytical and computational methods. Other 3D polyhedral geometries can be easily fabricated using 3D nanoprinting methods, such as two-photon lithography, to access a huge phase space of additional crystal phases, especially when under quasi-2D confinement. While the size of the current lattices is too large for optical frequency photonic crystals or metamaterials, two-photon lithographed structures can be shrunk up to ≈20% of their original size to form sub-micron scale particles through pyrolysis[65]. In addition, chemistries exist for directly printing high dielectric materials such as silica, which is also necessary for optical applications[66]. Magnetic,

plasmonic and luminescent nanoparticles can be incorporated into photoresists to impart further functionality and enable self-assembly under external stimuli. This could be used to generate programmable matter in which dynamic phase transitions are used to switch between structures and properties.

## Methods

### Fabrication of tetrahedrons and truncated tetrahedrons

Microscale tetrahedrons, truncated tetrahedrons ($t = 7/10$) and ATTs ($t = 2/3$) are fabricated using two-photon lithography on the Nanoscribe Photonic GT (Nanoscribe, GmbH). Three-dimensional models of tetrahedrons and ATTs are generated in Solidworks 2021 and then exported to STL files. These STL files are then imported into slicing software (DeScribe 2019, Nanoscribe, GmbH), to control printing conditions. The particles are printed in $10 \times 10$ arrays, resulting in a total of ≈50,000 particles for a single print. IP-Dip resist (Nanoscribe, GmbH), and a high magnification objective (63× NA 1.4 Zeiss) are used to fabricate the particles on a quartz coverslip (0.25 mm, SPI Supplies). After fabrication, the particles are developed in SU-8 developer (Kayaki Advanced Materials) for 10 min and then 2-propanol (> 99.5%, J.T. Baker) for 1 min. The particles are placed under a UV lamp for 30 min to improve surface roughness and cure any remaining surface monomers. The particles are treated with 1 w.t.% Pluronic F127 to stabilize the particles in solution. The substrate is then placed in a beaker filled with Milli-Q water and sonicated for < 30 s to remove the particles from the substrate. The solution is then transferred to a centrifuge tube and centrifuged at ≈7500 × $g$ for 20 min to aggregate the particles. The supernatant is removed, and the remaining solution is sonicated for 5–10 min to redisperse the particles.

### Colloidal assembly

The colloidal solution is deposited into a glass bottom well plate (Sensoplate, Greiner). The well plate is placed on an orbital shaker plate (Troemner Talboys, Fisher Scientific) at a setting of 5. This well plate is placed at an angle (≈10° for the tetrahedrons and ≈5° for the truncated tetrahedrons) to allow the particles to aggregate at the edge of the well plate. The particles sediment and assemble for several days (3–5 days) before imaging. To induce a phase transition, the particles are tilted at a higher angle for several days (3–5 days) before imaging.

### Microscopy

Bright-field optical images are captured using a Nikon Eclipse Ti2 with a CCD camera. Confocal images are taken using a Zeiss LSM 780 microscope. For high magnification images, an index matching oil is used between the objective and the glass bottom of the well plate. SEM images are taken on a FEI Helios NanoLab 600i Dual Beam SEM/FIB. For in-situ videos, particles are imaged over several hours using a Nikon Eclipse Ti2 with a CCD camera (0.2 frames per s).

### Monte Carlo simulations

Three-dimensional models of ATTs are generated in Solidworks 2021, and the vertex coordinates are referenced with the origin (0,0,0) coincident with the center of mass. HOOMD hard particle Monte Carlo package (v3.2.0) is used to simulate the assembly of ATTs. For the hexagonal structure, two impenetrable planes are placed at the top and bottom of the simulation box to constrain motion to a 2D plane (to prevent out-of-plane rotation). Particles ($N = 400$) are initialized in a simple array and Monte Carlo steps are run to randomize the initial configuration. After this, the simulation box is compressed in $x$-$y$. The final hexagonal phase is used as the initial configuration for the simulation of the quasi-diamond structure. The top impenetrable plane is raised to allow out-of-plane rotation. The simulation box is then compressed in all directions. These simulations are stopped once the simulation box dimensions converge, and the structure is stable.

## Truncation parameter

The truncation parameter describes the level of truncation of a tetrahedron. The truncation parameter, $t$, can range from 0 to 1, and corresponds to a regular tetrahedron when $t = 0$ and a regular octahedron when $t = 1$. A truncated tetrahedron with truncation parameter of $t$ will have 4 equilateral triangles with edge length $a(t/2)$ and four

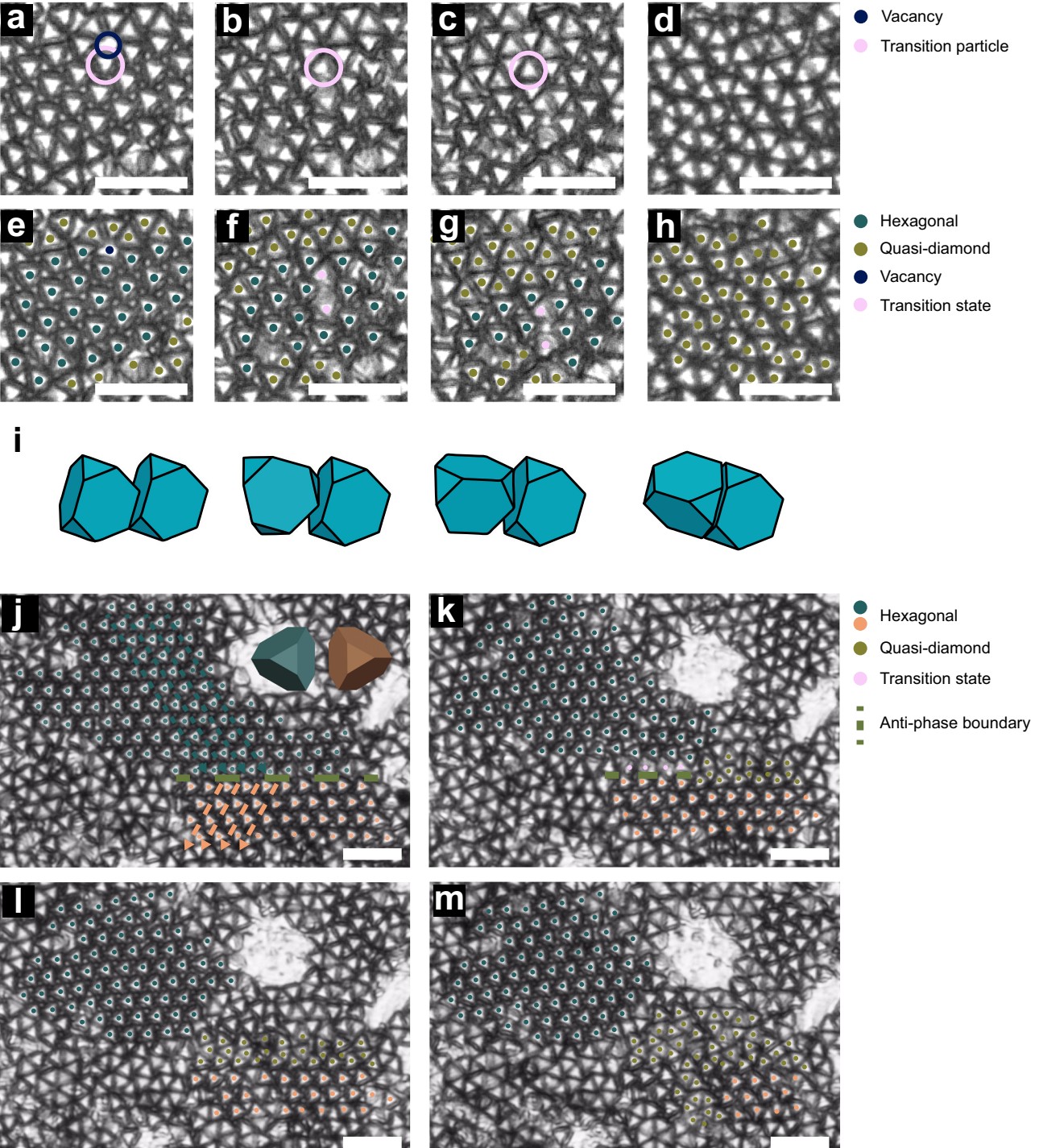

**Fig. 4 | Direct imaging of defect mediated phase transitions.** In-situ optical images of the (**a**) initial hexagonal grain and vacancy, (**b**) first particle rotation, (**c**) propagation of the phase transition through the hexagonal grain, (**d**) final quasi-diamond state. The vacancy is marked by a navy-blue circle. The adjacent ATT is marked by a pink circle, which is the first particle to transform. **e**–**h** The same images with colors that indicate the hexagonal phase (green), quasi-diamond phase (mustard yellow), particle in transition (pink), and vacancy (navy-blue). **i** Illustration of the kinetics of a particle rotation from an 'upright' to 'upside-down' position. **j** Two hexagonal grains with different orientations are shown in green and peach with corresponding 3D models. These grains are separated by an anti-phase boundary (army-green dashed line). Green and peach arrows show the alignment of the particles and point in the direction of a triangular vertex. **k** Transition of hexagonal grains (green or peach) to quasi-diamond (mustard yellow) at the anti-phase boundary is preceded by the rotation of particles into a transition state (pink) along these rows. **l** The anti-phase boundary is replaced by the quasi-diamond phase (mustard yellow) which separates the two remaining hexagonal grains (green or peach). **m** The phase transition begins to propagate in the lower grain and transform the hexagonal phase (peach) to the quasi-diamond phase (mustard yellow). All scale bars are 25 μm.

hexagons with two edge lengths of $a(1-t)$ and $a(t/2)$ as described by Damasceno et al.[49]

## Bond-order analysis

The bond orientational order parameter describes the angular positions of neighboring particles. The bond orientational order parameter, $\psi_{k,p}$, is defined as:

$$\psi_{k,p}^{a} = \frac{1}{p}\sum_{b} e^{ik\alpha_{ab}} \qquad (2)$$

where $a$ is the reference particle, $b$ is a neighboring particle, $k$ is the fold symmetry, $p$ is the number of expected neighboring particles, and $\alpha_{ab}$ is the angle between $a$ and $b$ in the global frame. This is calculated for neighboring particles, $b$, a certain radius away from the reference particle. This radius is equal to the first valley after the first peak in the $g(r)$. For a quasi-diamond phase, $k = p = 3$. For a hexagonal phase, $k = p = 6$. The resulting $\psi_{k,p}$ is a complex number that can be represented on a color wheel, in which the $x$-axis (real) is normalized to the average bond order, $\langle \psi_{k,p} \rangle$.

## Reporting summary

Further information on research design is available in the Nature Portfolio Reporting Summary linked to this article.

## Data availability

The data that support the findings of this study is deposited in Dryad[67] and are available from the corresponding author upon request.

## Code availability

Code for simulation and analysis are deposited in Dryad[67] and are available from the corresponding author upon request.

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

## Acknowledgements

We thank Prof. Matthew Jones for helpful advice on experiments and the manuscript. D.D. acknowledges the National Science Foundation Graduate Research Fellowship under Grant No. 1656518. J.K. is supported by a Stanford Graduate Fellowship. D.D., J.K., and X.W.G. acknowledge funding from the Hellman Foundation, and the National Science Foundation under Grant No. CMMI-2052251. Part of this work was performed at the Stanford Nano Shared Facilities (SNSF), which is supported by the National Science Foundation under award ECCS-1542152. Part of this work was performed at the Stanford Cell Sciences Imaging Facility.

## Author contributions

D.D. and J.K. performed experiments, analyzed data, and wrote the manuscript. D.D. performed Monte Carlo simulations. J.K. wrote codes for bond-order analysis. D.D., J.K., and X.W.G. commented on and edited the manuscript. X.W.G. oversaw the project.

## Competing interests

The authors declare no competing interests.
