## [Peer Review File · Nature Communications]

Direct observation of phase transitions in Archimedean truncated tetrahedrons under quasi-2D confinementREVIEWER COMMENTS

Reviewer #1 (Remarks to the Author):

Building on the accurate fabrication of batches of microscopic Archimedean truncated tetrahedra, this manuscript demonstrates their assembly into two different structures and provides an investigation of the transition from the low-density to the higher-density phase. The authors describe in depth the assembly conditions and the crystalline and defect structure of the ordered phases. In addition to the detailed experimental study, the authors provide corroborating evidence in the form of numerical simulations, as well as theoretical underpinning of the transition between these quasi-two-dimensional structures.

This study is well planned, researched, and presented and reports results significant to the field of condensed colloidal matter and the study of structural phase transformations.

A few minor comments should be addressed by the authors in a revised manuscript:

1. On line 77, the authors state that Archimedean truncated tetrahedra are predicted to form alpha-As at high packing densities. The cited paper [42], however, reports that the densest packing of these shapes is alpha-As, without stating that the assembly of that structure is observed or likely.
2. The authors use the term "hexatic" for the hexagonal phase that they observe at a 5° tilt of the substrate. Hexatic phases are anisotropic fluid phases, whereas the described phase seems to be solid, given that the authors even describe the particle positions as "locked in" due to their respective facet arrangement. The authors could consider calling their phase "hexagonal" instead.
3. The term "triatic", which the authors use on line 118, might not be necessary. It refers to liquid crystals specifically, but the here-discussed phase seems to be crystalline, and the authors don't use the term again throughout their manuscript but call the phase "quasi-diamond" instead.
4. On lines 126–127, the authors state that the diamond structure type was "predicted to be the lowest energy 3D structure of ATTs", however, the cited reference [42] merely reported the self-assembly of that phase without reporting the energies of this or alternative structures composed of ATT-shaped particles.
5. In the context of framing their free-energy calculations of packings of anisotropic particles, the authors should mention that other methods have been used to compute the free-energy landscapes of particles with faceted shapes (e.g., <https://www.pnas.org/doi/full/10.1073/pnas.1720139115>). Additionally, the authors describe the calculations of their own systems in relatively little detail, in particular compared with their very detailed description of the free-energy calculations of systems of spheres; more in-text description in addition to the depictions of the Voronoi cells in Fig. 3 would be helpful.

6. Three small inaccuracies among the references should be corrected:

- Reference 14: The article number is 801 (instead of the indicated page numbers 1–8).
- Reference 44: The article number is 33601 (instead of the indicated page numbers 1–12).
- Reference 45: The article was published in 2015 (instead of 2014).

7. The described experimental setup involves particles settled on the bottom of a plate, as opposed to the simulations in which particles are confined on both top and bottom. When the plate is tilted, what prevents particles from piling up out-of-plane? Is there a critical tilt angle at which the particles would no longer act quasi-2D?

Additionally, since the authors observe the hexagonal and quasi-diamond phases in systems of Archimedean truncated tetrahedra, and a triangular tiling in regular (Platonic) tetrahedra, it would be interesting to investigate the crossover point between these behaviors. What do tetrahedra at intermediate truncations (e.g., the space-filling truncated tetrahedra at $t = 0.5$) assemble?

Reviewer #2 (Remarks to the Author):

A. The authors investigate the self-assembly of Archimedean truncated tetrahedrons (ATT) in quasi-2D confinement. ATT microparticles were fabricated with high monodispersity using two-photon lithography and assembled on a tilted substrate. As the tilt angle increases, the system transitions from a hexatic phase, where all particles exhibit uniform orientations, to a quasi-diamond phase marked by a complementary arrangement wherein half the particles face upwards and the other half downwards. The study analyzes the thermodynamics and kinetics of these phase transitions, highlighting the role of defects in mediating the transitions by allowing out-of-plane particle rotation. The work also suggests the potential for creating various polyhedral geometries through 3D nanoprinting, with applications in photonics and metamaterials. The experiments and simulations were nicely conveyed and described clearly throughout the main text. I recommend the publication of this manuscript after careful revisions.

B. Some of the interpretations raise controversy. The most critical concern lies in whether the triggering of the solid-solid transformation is thermal or athermal. In Line 91, the authors claimed that the particles generally faced upwards after sedimentation on a flat plane. One thing that I want to verify is whether particles can spontaneously flip. The authors could provide a supplementary video showing the particles' dynamics in the fluid phase since their particles are ~ 5 μm in diameter and are synthesized with SU8, whose density is $\sim 1.1\rho_{\text{water}}$. I assume that spontaneous flipping due to thermal fluctuation could be rare. It is crucial to provide the audience with a basic sense of how significant the thermal fluctuation is in their system. Can the audience also estimate the energy barrier for a single particle to flip based on the density and geometry of their particles and compare it with kT ?

C. In the paragraph starting from line 140, the authors mainly discuss the entropy of hexatic and quasi-

diamond phase based on the free volume of their particles. They perform experiments with tilted samples under gravity. Gravity leads to a pressure and density gradient along the z direction depending on the tilt angle. The authors should clarify this point.

D. Can the authors draw 3D schematic diagrams showing how a particle flips relative to its neighbors during the solid-solid transition? It is hard for the audience to imagine the three-dimensional rotation from bright-field images in Fig. 4. There also seems to be drifting in Fig. 4(e-h). Can the authors align the pictures so that the area where the transition happens stays in the center of the image?

E. In the last part about the kinetics of the transition, does the transition proceed along the direction of the tilting or proceed randomly? Again, the focus is whether the phase transition is thermodynamically or mechanically driven. If gravity plays a key role, the flipping should have a preferred direction, which may lead to the solid-solid interface moving along the tilting direction. On the contrary, the interface should proceed more or less randomly if the entropy purely drives the phase transition.

Reviewer #3 (Remarks to the Author):

I have read the manuscript NCOMMS-23-34707-T by Wendy Gu et al. with interest. The authors have synthesized interesting particles using 2-photon lithography. However, to my mind, the results are not conclusive as the structures are probably not equilibrium structures as discussed in more detail below. The interpretation of the results is not convincing. The introduction is not accurate.

For example, in lines 33-34, the authors write “3D structures ... usually occur under complex interactions”. Well, most publications in the field of colloidal crystals deal with hard (or, nearly hard) spheres. What “complex interactions” are here? On lines 45-47 the authors state that “phase transitions studies ... have been limited to 2D ... that require complicated external fields.” Well, microscopy is easier in 2D, which explains a relatively large number of 2D studies. Still, plenty of 3D studies of structures and transitions between them use scattering techniques or confocal microscopy. I feel that the authors should do a better job of properly describing the scenery.

In the abstract and the text the authors talk about “small gravitational potential”. What does “small” mean here? I guess that the gravitational length of the particles is smaller than their size, isn't it? Please provide numbers! Also, how large is the gravitational energy that is needed to put particles upside down?

The assignment of the hexatic phase is not convincing. Hexatic structure is characterized by the presence of many dislocations that leave bond orientational order (quasi-)long range but destroys positional order. I would say that the structure shown in Fig. 1 is a poorly organized hexagonal structure with a small domain size. I do not see dislocations within the domains. The positional order seems well preserved within domains. Fig. 1e shows $g(r)$, which exhibits clear peaks that extend over distances larger than the

typical radius of the domains.

By the way, what is the polydispersity of the particles? If the authors like to continue searching for the hexatic phase, let me mention that it could also be induced by polydispersity, as observed for example in the columnar phase of colloidal platelets. Strain caused by the polydispersity could be released by dislocations, thus promoting the formation of the hexatic structure with long-range bond order but short positional order.

The details of the transition to the bilayer quasi-diamond structure (Fig. 4) are interesting. Still, is it a phase transition as the authors describe? Are the structures observed equilibrium structures? Would the bilayer transform back to a single layer if tilt is removed? My guess is no, no, and no. Moreover, Avrami's theory is not about equilibrium, is it?

To summarize, the investigation is potentially interesting but I see significant problems with this manuscript. I do not think it can be improved to meet the requirements of Nat Comm.

Reviewer #4 (Remarks to the Author):

RESPONSE TO REVIEWERS' COMMENTS

We would like to thank the reviewers for their valuable comments/suggestions. A detailed and point-to-point response to all concerns raised by the reviewers are listed below.

Reviewer #1 (Remarks to the Author):

Building on the accurate fabrication of batches of microscopic Archimedean truncated tetrahedra, this manuscript demonstrates their assembly into two different structures and provides an investigation of the transition from the low-density to the higher-density phase. The authors describe in depth the assembly conditions and the crystalline and defect structure of the ordered phases. In addition to the detailed experimental study, the authors provide corroborating evidence in the form of numerical simulations, as well as theoretical underpinning of the transition between these quasi-two-dimensional structures.

This study is well planned, researched, and presented and reports results significant to the field of condensed colloidal matter and the study of structural phase transformations.

A few minor comments should be addressed by the authors in a revised manuscript:

1. On line 77, the authors state that Archimedean truncated tetrahedra are predicted to form alpha-As at high packing densities. The cited paper [42], however, reports that the densest packing of these shapes is alpha-As, without stating that the assembly of that structure is observed or likely.

> We thank the reviewer for pointing this out. We have changed the sentence on line 76-78 to:

For example, ATTs ($t = 2/3$), which have four regular hexagonal faces and four regular triangular faces with all the same edge lengths, are predicted to form diamond structure at lower packing densities (~ 0.6), with α -arsenic as the densest packing structure (~ 1)⁴⁹.

2. The authors use the term "hexatic" for the hexagonal phase that they observe at a 5° tilt of the substrate. Hexatic phases are anisotropic fluid phases, whereas the described phase seems to be solid, given that the authors even describe the particle positions as "locked in" due to their respective facet arrangement. The authors could consider calling their phase "hexagonal" instead.

> We thank the reviewer for the comment, and have changed all mention of "hexatic" to "hexagonal" throughout the manuscript.

3. The term "triatric", which the authors use on line 118, might not be necessary. It refers to liquid crystals specifically, but the here-discussed phase seems to be crystalline, and the authors don't use the term again throughout their manuscript but call the phase "quasi-diamond" instead.

> We have removed the term triatic from line 118 and instead added (lines 118-119):

This results in a phase that is drastically different than the previous hexagonal phase.

4. On lines 126–127, the authors state that the diamond structure type was "predicted to be the lowest energy 3D structure of ATTs", however, the cited reference [42] merely reported the self-assembly of that phase without reporting the energies of this or alternative structures composed of ATT-shaped particles.

> We thank the reviewer for the comment. As pointed out, reference [42] does not specify the exact energies or alternative structures composed of ATT-shaped particles. We have changed this in line 127-128:

This has been predicted to form from ATTs that self-assemble under an entropic driving force at packing fractions above 0.50⁴⁹.

5. In the context of framing their free-energy calculations of packings of anisotropic particles, the authors should mention that other methods have been used to compute the free-energy landscapes of particles with faceted shapes (e.g., <https://www.pnas.org/doi/full/10.1073/pnas.1720139115>). Additionally, the authors describe the calculations of their own systems in relatively little detail, in particular compared with their very detailed description of the free-energy calculations of systems of spheres; more in-text description in addition to the depictions of the Voronoi cells in Fig. 3 would be helpful.

> We would like to thank the reviewer for bringing this work to our attention. We have cited it on lines 164-166:

Other similar cell methods have also been developed to calculate the densest packing states of polygons, and subsequently, the free volume of certain packing structures^{61,62}.

> We have also increased the description of the single occupancy cells that are generated to calculate the free volume of each phase and added an additional supplemental figure (see Supplementary Fig. S4 below) to describe the method on lines 181-187:

*Instead of constructing Voronoi cells and dilating in 3D, self-similar cells are constructed around each particle from their phase (either hexagonal or quasi-diamond) and then dilated in two dimensions (**Figure 3a**) (also see Supplementary Information). That is, the single-cell is constructed to be in a shape of an ATT, and then stretched only in the x-y dimension. Because of the shape of the cell, the volume of the single cell, V_{cell} , can be calculated analytically for different dilated states. The accessible free volume, V_{free} , is then taken as $V_{cell} - V_{ATT}$, where V_{cell} varies as a function of the dilation and V_{ATT} is constant.*

Supplementary Fig. 4. Single-cell occupancy model. Model of an ATT (teal) in a single particle, hexagonal, and quasi-diamond structure, with a self-similar (transparent) cell constructed around it at (a) the highest packing fraction, (b) intermediate packing fraction, and (c) lowest packing fraction. The free volume is the volume that is available for the ATT to move within its self-similar cell.

6. Three small inaccuracies among the references should be corrected:

- Reference 14: The article number is 801 (instead of the indicated page numbers 1–8).
- Reference 44: The article number is 33601 (instead of the indicated page numbers 1–12).
- Reference 45: The article was published in 2015 (instead of 2014).

> We thank the reviewer for pointing out the errors in our citations. We have fixed these.

7. The described experimental setup involves particles settled on the bottom of a plate, as opposed to the simulations in which particles are confined on both top and bottom. When the plate is tilted, what prevents particles from piling up out-of-plane? Is there a critical tilt angle at which the particles would no longer act quasi-2D?

> We first allow the particles to sediment on the substrate at very low packing fractions. The substrate area is much larger than the projected area of the particles. This means it is very unlikely that particles are stacked on top of each other at this stage (we have never observed this). After this, the substrate is tilted to allow the particles to assemble at the edge of the glass well as described in lines 85-97. Multiple layers are observed at the lowest side of the well. Interesting structures may form in the multi-layered region, but we are not able to image these structures because it is difficult to image through multiple layers (see image below). We generally only see multi-layered regions at the edge of the well at a tilt angle of ~10 degrees or greater. The formation of this multi-layered region depends on both particle packing density (distance between neighboring particles), and osmotic pressure (pressure exerted by other particles).

Additionally, since the authors observe the hexagonal and quasi-diamond phases in systems of Archimedean truncated tetrahedra, and a triangular tiling in regular (Platonic) tetrahedra, it would be interesting to investigate the crossover point between these behaviors. What do tetrahedra at intermediate truncations (e.g., the space-filling truncated tetrahedra at $t = 0.5$) assemble?

> We thank the reviewer for the good question. We expect tetrahedra with intermediate truncations (e.g., $t=0.5$) to form the triangular tiling. Only the Archimedean truncated tetrahedron can accommodate the quasi-2D to 3D assembly at an interface. To form the 2D assembly, the particle needs to have a preferential orientation during low tilt conditions (i.e. a low center of gravity weighted towards one face) and needs to be able to assemble in this orientation. Then, the particle needs to form a 3D assembly that can be flat against a 2D interface. To form a space-filling 3D assembly against this planar interface, the truncated tetrahedron must have equilateral polygon faces. Out of the truncated tetrahedra family, all of these conditions are only satisfied by Archimedean truncated tetrahedra. This has been observed in simulations of truncated tetrahedra, in which assembled ATTs can form against a flat interface, but lower truncations cannot form flat interfaces (Skye et al. *Soft Matter*, 2022, 18, 6782).

Reviewer #2 (Remarks to the Author):

A. The authors investigate the self-assembly of Archimedean truncated tetrahedrons (ATT) in quasi-2D confinement. ATT microparticles were fabricated with high monodispersity using two-photon lithography and assembled on a tilted substrate. As the tilt angle increases, the system transitions from a hexatic phase, where all particles exhibit uniform orientations, to a quasi-diamond phase marked by a complementary arrangement wherein half the particles face upwards and the other half downwards. The study analyzes the thermodynamics and kinetics of these phase transitions, highlighting the role of defects in mediating the transitions by allowing out-of-plane particle rotation. The work also suggests the potential for creating various polyhedral geometries through 3D nanoprinting, with applications in photonics and metamaterials. The experiments and simulations were nicely conveyed and described clearly throughout the main text. I recommend the publication of this manuscript after careful revisions.

B. Some of the interpretations raise controversy. The most critical concern lies in whether the triggering of the solid-solid transformation is thermal or athermal. In Line 91, the authors claimed that the particles generally faced upwards after sedimentation on a flat plane. One thing that I want to verify is whether particles can spontaneously flip. The authors could provide a supplementary video showing the particles' dynamics in the fluid phase since their particles are ~ 5 μm in diameter and are synthesized with SU8, whose density is $\sim 1.1\rho_{\text{water}}$. I assume that spontaneous flipping due to thermal fluctuation could be rare. It is crucial to provide the audience with a basic sense of how significant the thermal fluctuation is in their system. Can the audience also estimate the energy barrier for a single particle to flip based on the density and geometry of their particles and compare it with kT ?

> The transformation is athermal. Our group previously investigated the rotation of two-photon lithographed colloidal particles made out of the same material as in the current study (Doan et al. *Part. Part. Syst. Charact.* 2021, 38, 2100033). Out-of-plane rotation was not observed for tetrahedra with volumes of $\sim 60 \mu\text{m}^3$, while out-of-plane rotation was observed for tetrahedrons with volumes of $\sim 10 \mu\text{m}^3$. Proximity of the particle to the substrate hinders the rotation of the particle. The distance of the particle from the substrate is determined by the gravitational potential of the particle (which depends on its mass) and the repulsion from the substrate. As the volume of the particle increases, the particle is closer to the substrate, and rotation is more unlikely.

The volume of an ATT particle in the current manuscript is $\sim 113 \mu\text{m}^3$. This is twice the volume of the tetrahedra that did not experience out-of-plane rotation, which suggests that out-of-plane rotation in the ATTs is very rare. We have performed additional experiments to obtain the supplemental video suggested by the reviewer, and conclude that the ATTs do not rotate over long periods of time. We have attached optical videos of ATTs in solution at 0-degree tilt (no rotation), as well as confocal videos of tetrahedrons of volumes $\sim 10 \mu\text{m}^3$ (rotation occurs) and $60 \mu\text{m}^3$ (no rotation) (Video R1 and Video R2).

The flipping of a particle corresponds to an increase in gravitational potential energy. The gravitational energy required to flip a particle upside down is calculated as $\Delta E = mg\Delta h$, where Δh is the change in height of the center of gravity of the particle in its "upright" (hexagonal face is adjacent to the substrate) vs "upside down" (triangular face is adjacent to the substrate) position. The energy needed to flip the

particles is $\sim 10 k_B T$, which suggests that it is very unlikely that out of plane rotation can occur spontaneously at room temperature.

This discussion has been added to line 198-206:

The flipping of a particle corresponds to an increase in gravitational potential energy. The gravitational energy required to flip a particle upside down is calculated as $\Delta E = mg\Delta h$, where Δh is the change in height of the center of gravity of the particle in its “upright” (hexagonal face is adjacent to the substrate) vs “upside down” (triangular face is adjacent to the substrate) position. The energy needed to flip the particles is $\sim 10 k_B T$, which suggests that it is very unlikely that out of plane rotation can occur spontaneously at room temperature without external energy input (see Movie S1). In addition, a kinetic barrier also exists due to the free volume required to mediate the rotation of a particle out-of-plane from its locked hexagonal phase to the quasi-diamond phase.

C. In the paragraph starting from line 140, the authors mainly discuss the entropy of hexatic and quasi-diamond phase based on the free volume of their particles. They perform experiments with tilted samples under gravity. Gravity leads to a pressure and density gradient along the z direction depending on the tilt angle. The authors should clarify this point.

> The reviewer makes a good point. We clarify this point in lines 94-96:

The substrate is then tilted to apply an in-plane (x/y direction) gravitational potential field. This gravitational field leads to an induced osmotic pressure and density gradient along the direction of tilt.

D. Can the authors draw 3D schematic diagrams showing how a particle flips relative to its neighbors during the solid-solid transition? It is hard for the audience to imagine the three-dimensional rotation from bright-field images in Fig. 4. There also seems to be drifting in Fig. 4(e-h). Can the authors align the pictures so that the area where the transition happens stays in the center of the image?

> We have added this schematic to Figure 4 (as (i)) and below, and have referenced it in the main text in line 229-231.

The hexagonal particle adjacent to the vacancy rotates out-of-plane and into an ‘upside-down’ position, in which a triangular face is adjacent to the substrate (Figure 4i).

For Figure 4(e-h), we have centered each image around the grain of interest using a tracking algorithm that accounts for drift. There is no drift (the same particles are in the frame through the video). The apparent motion is due to the motion of particles during the phase transition.

(i) Illustration of particle rotation.

E. In the last part about the kinetics of the transition, does the transition proceed along the direction of the tilting or proceed randomly? Again, the focus is whether the phase transition is thermodynamically or mechanically driven. If gravity plays a key role, the flipping should have a preferred direction, which may lead to the solid-solid interface moving along the tilting direction. On the contrary, the interface should proceed more or less randomly if the entropy purely drives the phase transition.

> For the images in Figure 4 of the manuscript, the propagation of the phase transformation is perpendicular to the tilt direction. The transformation starts with flipping of one particle, which leads to flipping of additional particles. However, this is not always the case. We performed additional experiments (see attached Video R3) that shows a grain that transforms more randomly. In this video, the grain of interest has multiple flipping events around its grain boundary, which all propagate inward to transform the grain to the quasi-diamond phase. This shows that there is a delicate balance between mechanical and entropic driving forces in the system. Mechanical driving forces are likely to dominate due to the large volume of the particles, but entropy also plays a role.

This has been added to the manuscript on lines 247-253:

In Figure 4, the propagation of the phase transformation is perpendicular to the tilt direction with a particle flipping the transition proceeding in a linear direction. However, this is not always the case. Supplementary Movie 5 shows a randomly proceeding phase transition for ATTs. Multiple particle flipping events occur and propagate inward, transforming the structure from a hexagonal to quasi-diamond phase. This suggests that while a mechanical driving force is necessary to induce the phase transition, entropy does in fact play a role. Likely, the phase transition is driven by a combination of mechanical and thermodynamic driving forces.

Reviewer #3 (Remarks to the Author):

I have read the manuscript NCOMMS-23-34707-T by Wendy Gu et al. with interest. The authors have synthesized interesting particles using 2photon lithography. However, to my mind, the results are not conclusive as the structures are probably not equilibrium structures as discussed in more detail below. The interpretation of the results is not convincing. The introduction is not accurate.

For example, in lines 33-34, the authors write “3D structures ... usually occur under complex interactions”. Well, most publications in the field of colloidal crystals deal with hard (or, nearly hard) spheres. What “complex interactions” are here? On lines 45-47 the authors state that “phase transitions studies ... have been limited to 2D ... that require complicated external fields.” Well, microscopy is easier in 2D, which explains a relatively large number of 2D studies. Still, plenty of 3D studies of structures and transitions between them use scattering techniques or confocal microscopy. I feel that the authors should do a better job of properly describing the scenery.

> We apologize for the unclear statements. We have edited lines 28-33 to read:

Hard or nearly hard spheres are commonly observed to form face-centered cubic structures. Complex three-dimensional (3D) structures such as diamond, space-filling polyhedral packing, and porous lattices have been formed by using patchy DNA interactions, shape-dependent entropic forces, or magnetic, gravitational, and capillary forces¹⁷⁻²². A wide range of hard-particle 3D assemblies have been predicted to form from polyhedra in simulation²³, but are challenging to experimentally achieve.

An additional recent review paper has been added to the citations (Li et al. *Chemical Reviews*. 2022, 122, 4976-5067).

We have edited lines 43-45 to include 3D studies and include additional references:

The majority of the studies have been in 2D systems that require complicated external fields. 3D phase transitions (e.g. from FCC phase to AuCu phase) have been observed using X-ray scattering³²⁻³⁴ and confocal techniques³⁵⁻³⁷.

In the abstract and the text the authors talk about “small gravitational potential”. What does “small” mean here? I guess that the gravitational length of the particles is smaller than their size, isn't it? Please provide numbers! Also, how large is the gravitational energy that is needed to put particles upside down?

> We apologize for the ambiguous adjective and have removed it from the abstract and line 94. Instead, we have modified the abstract sentence to say:

“These particles self-assemble into a hexagonal phase under an in-plane gravitational potential.”

We compute the gravitational length to be $k_B T / mg \sin(5 \text{ degrees})$, where $m = \Delta\rho \cdot V_{ATT}$, and $\Delta\rho$ is the difference in density between the polymer and the water ($\sim 0.1 \text{ g/cm}^3$). This results in a gravitational length of $\sim 1.9 \text{ um}$.

The flipping of a particle corresponds to an increase in gravitational potential energy. The gravitational energy required to flip a particle upside down is calculated as $\Delta E = mg\Delta h$, where Δh is the change in height of the center of gravity of the particle in its “upright” (hexagon face adjacent to substrate) vs “upside down” (triangular face adjacent to substrate) position. The energy needed to flip the particles is $\sim 10 k_B T$, which suggests that it is very unlikely that out of plane rotation can occur spontaneously at room temperature.

The assignment of the hexatic phase is not convincing. Hexatic structure is characterized by the presence of many dislocations that leave bond orientational order (quasi-)long range but destroys positional order. I would say that the structure shown in Fig. 1 is a poorly organized hexagonal structure with a small domain size. I do not see dislocations within the domains. The positional order seems well preserved within domains. Fig. 1e shows $g(r)$, which exhibits clear peaks that extend over distances larger than the typical radius of the domains.

> We thank the reviewer for the comment. We have changed “hexatic” to “hexagonal” throughout the manuscript. The focus of our manuscript is that we have an interesting phase formation and transition not previously seen in literature. These types of assemblies have mainly been predicted in simulation, and not been easily seen experimentally. Our ability to fabricate complex particle shapes allows us to unlock new directions for colloidal studies.

By the way, what is the polydispersity of the particles? If the authors like to continue searching for the hexatic phase, let me mention that it could also be induced by polydispersity, as observed for example in the columnar phase of colloidal platelets. Strain caused by the polydispersity could be released by dislocations, thus promoting the formation of the hexatic structure with long-range bond order but short positional order.

> Previous computational and experimental studies shown that polydispersity greater than $\sim 7\%$ of hard particle spheres can completely disrupt nucleation and crystallization, and could be used to purposely induce a hexatic phase (Pusey et al. *Phil. Trans. R. Soc. A.* 2009, 367, 4993-5011, Zaccarelli et al. *PRL.* 2009, 103, 135704, Gasset et al. *J. Phys.:Condens. Matter.* 2009, 21, 203101). In our study, the particle size has a standard deviation of no more than $\sim 3\%$, which was characterized in a previous study from our group (Doan et al. *Part. Part. Syst. Charact.* 2021, 38, 2100033) Therefore, polydispersity is unlikely to have a large influence on the self-assembled structure.

The details of the transition to the bilayer quasi-diamond structure (Fig. 4) are interesting. Still, is it a phase transition as the authors describe? Are the structures observed equilibrium structures? Would the bilayer transform back to a single layer if tilt is removed? My guess is no, no, and no. Moreover, Avrami's theory is not about equilibrium, is it?

> The hexagonal structure is a metastable phase that is stable for long time scales, and is not affected by small perturbations due to thermal fluctuations. The diamond structure is the equilibrium state. This is according to both the free energy calculations and experimental observations. Yet, we argue that the transformation from a metastable to an equilibrium structure is still a phase transformation. In Anderson

et al. *Nature*, 2002, *416*, 811-815, phase transitions for colloids have been defined as a change between a gas, liquid, solid and liquid crystalline phases. These phase transitions includes transitions from undercooled, supersaturated states and metastable states. In fact, these metastable transitions show some of the most interesting and rich dynamic behavior in colloidal science (Rouwhorst et al. *Nat Commun*, 2020, *11*, 3558, Stopper et al. *Phys. Rev. E*, 2018, *97*, 062602).

If these self-assembled structure are disturbed (e.g. sonicated) or the particles are redispersed the particles, the particles will reform into a hexagonal phase under low tilt, and transform back into quasi-diamond under high tilt conditions. However, after the formation of a quasi-diamond phase, untilting the sample does not return back to its hexagonal state. This shows that the hexagonal phase is a metastable state. This does not invalidate the phase transition that is described in this manuscript.

We use the Avrami equation to calculate the rate of change from one phase to another through *in-situ* video. The reviewer is correct that the Avrami equation is not about equilibrium, but this does not invalidate its use in our manuscript.

Reviewer #4 (Remarks to the Author):

REVIEWERS' COMMENTS

Reviewer #1 (Remarks to the Author):

The authors provide thorough edits of the manuscript and responses to reviewer comments. I gladly recommend publication of the manuscript in its current form.

(One small copyediting comment: on lines 192–193, the authors probably meant to remove either "goes asymptotically" or "approaches" from the sentence.)

Reviewer #2 (Remarks to the Author):

The authors have answered all my questions, and I am satisfied with the revisions. I have no further questions and recommend the publication of this innovative and interesting work.

Reviewer #3 (Remarks to the Author):

First of all, I'd like to apologize for the review delay due to the end-of-the-year rush and some health issues.

The authors have clarified several points and I would like to thank them for that. Still, one essential statement is not convincing: whether the observed phases are thermodynamically stable, metastable, or arrested states.

In particular, the authors refer to the review of Anderson and Lekkerkerker in *Nature* (2002), which is about the phase transitions between thermodynamically-driven transitions between different equilibrium phases. The authors argue that the hexagonal structure is metastable while the quasi-diamond structure is the stable one. Do the authors have proof of that?

The provided results of Monte Carlo simulations (Movie S2, not S1 as stated in the text, isn't it? Also check what you refer to as Movie S2 and S3 on line 212) are performed upon shrinkage of the simulation box. What happens if one goes back to the original box size? I would expect that at the packing fraction as low as 0.1 the fluid phase must be recovered again (unless the particles are sticky). By the way, is the gravitational energy for particle flipping included in the simulations? It does not seem to be the case. As I stated already in my first review, the results are interesting. I have also observed that the other referees are more positive about this work. Perhaps, as a compromise solution, could the authors add a definition of what exactly they mean by "phase transition" in this context? Are we talking about the states at (infinitely) high osmotic compression only? I hope it will not take much effort and time. With this addition and clarification of the role of gravity in MC simulations, and in view of the recommendations of other referees, I will be happy to recommend this manuscript for Nat Comm.

Andrei Petukhov

Reviewer #4 (Remarks to the Author):

REVIEWERS' COMMENTS

Reviewer #1 (Remarks to the Author):

The authors provide thorough edits of the manuscript and responses to reviewer comments. I gladly recommend publication of the manuscript in its current form.

(One small copyediting comment: on lines 192–193, the authors probably meant to remove either "goes asymptotically" or "approaches" from the sentence.)

> We thank the reviewer for their time and have edited the lines accordingly.

Reviewer #2 (Remarks to the Author):

The authors have answered all my questions, and I am satisfied with the revisions. I have no further questions and recommend the publication of this innovative and interesting work.

> We thank the reviewer for their time.

Reviewer #3 (Remarks to the Author):

First of all, I'd like to apologize for the review delay due to the end-of-the-year rush and some health issues.

The authors have clarified several points and I would like to thank them for that. Still, one essential statement is not convincing: whether the observed phases are thermodynamically stable, metastable, or arrested states.

In particular, the authors refer to the review of Anderson and Lekkerkerker in Nature (2002), which is about the phase transitions between thermodynamically-driven transitions between different equilibrium phases. The authors argue that the hexagonal structure is metastable while the quasi-diamond structure is the stable one. Do the authors have proof of that?

The provided results of Monte Carlo simulations (Movie S2, not S1 as stated in the text, isn't it? Also check what you refer to as Movie S2 and S3 on line 212) are performed upon shrinkage of the simulation box. What happens if one goes back to the original box size? I would expect that at the packing fraction as low as 0.1 the fluid phase must be recovered again (unless the particles are sticky). By the way, is the gravitational energy for particle flipping included in the simulations? It does not seem to be the case.

As I stated already in my first review, the results are interesting. I have also observed that the other referees are more positive about this work.

Perhaps, as a compromise solution, could the authors add a definition of what exactly they mean by "phase transition" in this context? Are we talking about the states at (infinitely) high osmotic compression only? I hope it will not take much effort and time. With this addition and clarification of the role of gravity in MC simulations, and in view of the recommendations of

other referees, I will be happy to recommend this manuscript for Nat Comm.
Andrei Petukhov

> We thank the reviewer for their detailed response. The “stable” state that we refer to is the state that is reached under higher osmotic pressures, as seen in simulations in literature. In this case, the stable structure that we are referring to is the quasi-diamond state.

We define phase transition as one where there is a distinct change in structure and/or equation of state (*J. Phys. Chem. B* 2019, 123, 42, 9038–9043, *PRL* 107.15 (2011): 155704) Practically, we see this structural change in $g(r)$ spacing and bond order as shown in Figure 1 and 2. The equation of state change can be seen when the system transitions between the hexagonal and quasi-diamond state, making an abrupt change in energy as a function of packing fraction. We show this analytically in Figure 3 using the single cell occupancy model. This can be visually observed in the simulations when the particles switch from a 2D-confined state to a non-2D confined state (the particles have more room to move once there is particle rotates).

For the Monte Carlo simulations, we did not implement gravitational field in the movies. Rather, we make the claim that an equivalent potential is employed by confining the particles between two planes. We provided evidence that this is the case by showing images of particles of a certain mass not being able to rotate out of plane.

One limitation of using gravitational field only is that the particles do not “feel” the osmotic pressure gradient induced by particles stacking up on each other at our 15-degree tilt angle condition, which may affect the accuracy of the simulation and our experiments.

Reviewer #4 (Remarks to the Author):
